### Subject Category:
Biochemistry, cellular and molecular biology

### Subject Areas:
biotechnology/health and disease and epidemiology

### Keywords:
cytokine storms, pyroptosis, pseudorabies virus

### Author for correspondence:
Shanshan Liu
e-mail: huojianbaifenbai@163.com

[†]Those authors contributed equally to this work.

# Cytokine storms and pyroptosis are primarily responsible for the rapid death of mice infected with pseudorabies virus

Wei Sun[1,†], Shanshan Liu[1,2,†], Xuefei Huang[1], Rui Yuan[1,2] and Jiansheng Yu[1,2]

[1]College of Agriculture, Tongren Polytechnic College, Bijiang District, Tongren City, Guizhou 554300, People's Republic of China
[2]National and Local Engineering Research Centre for Separation and Purification Ethnic Chinese Veterinary Herbs, Tongren City, Guizhou 554300, People's Republic of China

WS, 0000-0002-7958-6117; SL, 0000-0002-9388-9333

Pseudorabies virus (PRV), the causative agent of Aujeszky's disease, is one of the most harmful pathogens to the pig industry. PRV can infect and kill a variety of mammals. Nevertheless, the underlying pathogenesis related to PRV is still unclear. This study aims to investigate the pathogenesis induced by PRV in a mouse model. The mice infected with the PRV-HLJ strain developed severe clinical manifestations at 36 h post-infection (hpi), and mortality occurred within 48–72 hpi. Hematoxylin-eosin staining and qRT-PCR methods were used to detect the pathological damage and expression of cytokines related to an immune reaction in brain tissue, respectively. The cytokine storms caused by IFN-$\alpha$, IFN-$\beta$, TNF-$\alpha$, IL-1$\beta$, IL-6 and IL-18 were related to the histopathological changes induced by PRV. This pattern of cytokine secretion depicts an image of typical cytokine storms, characterized by dysregulated secretion of pro-inflammatory cytokines and imbalanced pro-inflammatory and anti-inflammatory responses. In addition, the pyroptosis pathway was also activated by PRV by elevating the expression levels of nod-like receptor protein 3, Caspase-1, Gasdermin-D and interleukin-1$\beta$/18. These findings provide a way for further understanding the molecular basis in PRV pathogenesis.

## 1. Introduction

Pseudorabies virus (PRV) or suid $\alpha$ herpesvirus 1 is the pathogen of porcine Aujeszky's disease (AD) which affects the respiratory

system, nervous system and reproductive systems [1]. Many mammals other than pigs, are susceptible to PRV infection, such as cattle, sheep, rabbits, cats, dogs, guinea pigs, rats and mice [2]. However, pigs are the only susceptible animals that can survive, although the prognosis of the disease largely depends on factors including inoculation site, virus strain and titer as well as the age of pigs [3,4]. AD is a highly infectious disease with high mortality in piglets. Transmission mainly occurs through direct contact with oral and nasal secretions but can also occur through aerosol and the placenta or sexual intercourse [1]. Therefore, the prevalence of PRV has led to a wide range of economic losses in the pork production industry. Inactivated and attenuated vaccines have been developed to delay or reduce swine death. However, they cannot eradicate the disease because none can prevent virus potential infection and reactivation and shedding of the virulent fields [5]. Due to the impact of AD on the pig industry, some countries are trying to eradicate AD based on the differentiating infected from vaccinated animals program. However, since 2011, the outbreak of AD occurred in pigs vaccinated with PR vaccine in China, which indicates that the AD vaccine cannot provide effective treatment to prevent wild PR infection [6].

Mice and rabbits are usually used to study PRV in the laboratory. After infection, animals showed abnormal excitement and nasal itching, accompanied by convulsions and rapid death. In mice, PRV almost manifested as a neurogenic infection of the central nervous system (CNS), accompanied by fulminant central nervous symptoms and high mortality [7]. PRV is known to cause severe encephalitis in piglets and various non-native hosts, even in humans [8]. Few studies have focused on the pathogenesis of encephalitis. It is known that pyroptosis is involved in the immune response in various types of cells, which can be triggered by various pathological stimuli, leading to the secretion of pro-inflammatory cytokines and intracellular contents [9]. Inflammation is a double-edged sword, which has a crucial role in metabolism. A mild inflammatory response could protect the body to a certain degree, help to repair damaged tissue and be beneficial to steady-state reconstruction. However, excessive inflammation may form 'cytokine storms', leading to tissue damage. In the present work, we describe the influence of PRV on the immune factor and pyroptosis-related factors in mice brains.

# 2. Material and methods

## 2.1. Reagents and animals

An Annexin V-FITC/PI Apoptosis kit was obtained from BD Company (Franklin Lakes, USA). Modified Bradford Protein Assay Kit, Antibody against NLRP3, Animal Total RNA Isolation Kit and Tissue Total Protein Extraction Kit were supplied by Sagon Biotech Company (Shanghai, China). Another antibody against Gasdermin D (GSDMD) was bought from Thermo Fisher Company (USA). IL-1$\beta$, IL-18 and $\beta$-actin antibodies were obtained from Bioss Company (Beijing, China). Caspase-1 antibody was obtained from Boster Biological Technology com.Ltd (Wuhan, Hubei, China). A PrimeScript RT reagent kit was bought from TAKARA company (Daliang, China). PRV-HLJ strain (MK080279.1) is a strain isolated from Heilong Jiang prpvince and provided by Professor Jingfei Wang from Harbin Veterinary Research Institute, CAAS. Eighty-six-week-old female Balb/C mice were obtained from Dossy Experimental Animal Corporation (Chengdu, China).

## 2.2. Experimental

One week later, the mice were divided randomly into five groups with 16 per group, including one control group and four experimental groups, in which mice were infected at 36 hpi, 48 hpi, 60 hpi and 72 hpi. The mice in the control group were injected with 0.2 ml of normal saline by subcutaneous inoculation on the back. The mice in another group received 0.2 ml PRV-HLJ strain ($10^4$ TCID$_{50}$/100 µl) at the same inoculation site. The animals were fed in the room illuminated with a 12 h light-dark cycle. The ambient temperature and relative humidity were maintained at 22–24°C and 40–60%, respectively. Ten mice were fed in each cage and given the above dosage. Water and diet were provided ad libitum. Mice brain tissues in each group were collected aseptically. Subsection specimens were snap-frozen and stored at −80°C for RNA extraction. In addition, portions of the brain were fixed in 4% paraformaldehyde solution for histopathological examination.

## 2.3. Histopathological analysis

The histopathological observation was operated by using a standard laboratory procedure. The brain was removed from experimental animals and washed thoroughly in phosphate-buffered saline (PBS, pH 7.4).

Then, the tissue was fixed in 4% paraformaldehyde for 2 d and transferred to 95% or absolute alcohol for dehydration. After that, it was processed to a paraffin embedding routine. The paraffin-embedded tissue was sliced into a 5 µm section, dewaxed in xylene and then rehydrated in graded alcohols. The section was stained with hematoxylin-eosin (H&E) and then examined under a light microscope for histopathological examination.

## 2.4. Detection of pyroptosis by flow cytometry in brain

The rate of pyroptosis cells in the brain was measured using an Annexin V-FITC/PI Apoptosis kit according to the instructions provided. Brains were taken from mice, which were humanely killed at the time mentioned above, ground to form a suspension and filtered with a 300-mesh nylon screen. The cells were washed three times with pre-cold PBS and adjusted at a concentration of $1 \times 10^6$ cells ml$^{-1}$. Furthermore, 100 µl cells were incubated with Annexin V-FITC/PI staining at room temperature for 15 min in a culture tube in a dark atmosphere. To each tube was added 300 µl of binding buffer and then detected with an FCM (Becton Dickinson, USA). CellQuest Pro software (Becton Dickinson, USA) was used to visualize the results.

## 2.5. RNA extraction and qRT-PCR analysis

Total RNA from brains was isolated by an 'Animal Total RNA Isolation Kit' according to the kit instructions. RNA integrity was detected by 2% agarose gel electrophoresis. A spectrophotometer was used to detect the RNA quantity and quality (NanoDrop-2000, Thermofisher Company, USA). Total RNA was converted into DNA for qRT-PCR by using the PrimeScript RT reagent kit. The first strand of cDNA was amplified through SYBR staining on a LightCycler 96 apparatus (Roche, Germany). All primers used in this research were designed by Oligo.7 software and synthesized by Sagon Biotech Ltd. (Shanghai, China). Detailed information about primers was available in the electronic supplementary material, table S1. The $2^{-\Delta\Delta Ct}$ method was used to analyse the mRNA expression. $\beta$-actin, a housekeeping gene, was used as an internal control for values correction.

## 2.6. Western blotting

A 'Tissue Total Protein Extraction Kit' was used for protein extracted from the brain. The protein concentration of each specimen was measured by Bradford assay. The proteins were first separated on a 10% sodium dodecyl sulfate-polyacrylamide gel electrophoresis (SDS-PAGE) and then transferred onto a polyvinylidene difluoride (PVDF) membrane. Next, the PVDF membrane was blocked with TBST solution containing 5% non-fat milk at room condition for 2 h and then incubated at 4°C condition for 12 h with the corresponding primary antibodies diluted with a solution: NLRP3 (1 : 1000), Caspase-1 (1 : 800), GSDMD (1 : 900), IL-1$\beta$ (1 : 500), IL-1$\beta$ (1 : 500) and $\beta$-action (1 : 2000). Then, the PVDF membrane was incubated at room temperature with HRP-labelled secondary antibody for 1 h, and blots were measured by an ECL reagent. The $\beta$-actin was used as a protein loading control.

## 2.7. Statistical analysis

SPSS22.0 software was used for data analysis. The results were presented as means ± standards deviations (mean ± s.d.). One-way analysis of variance package in SPSS 22.0 was used to evaluate the statistical significance between PRV and control group. GraphPad Prism6.0 software was used to perform statistical artwork. In all statistical comparisons, the $p$-value was introduced as a judgement of the statistical difference.

# 3. Results

## 3.1. Clinical symptoms and histopathological analysis

At 36 hpi, the mice infected with PRV generally appeared with typical clinical signs, including depression, anorexia and neuropathic itch. Mortality occurred within the period of 48–72 hpi. However, none of the mice in the control group exhibited clinical symptoms or died. Compared with the control group, there was a significant difference in the microscopic lesions in the PRV-inoculated group (figure 1). Normal

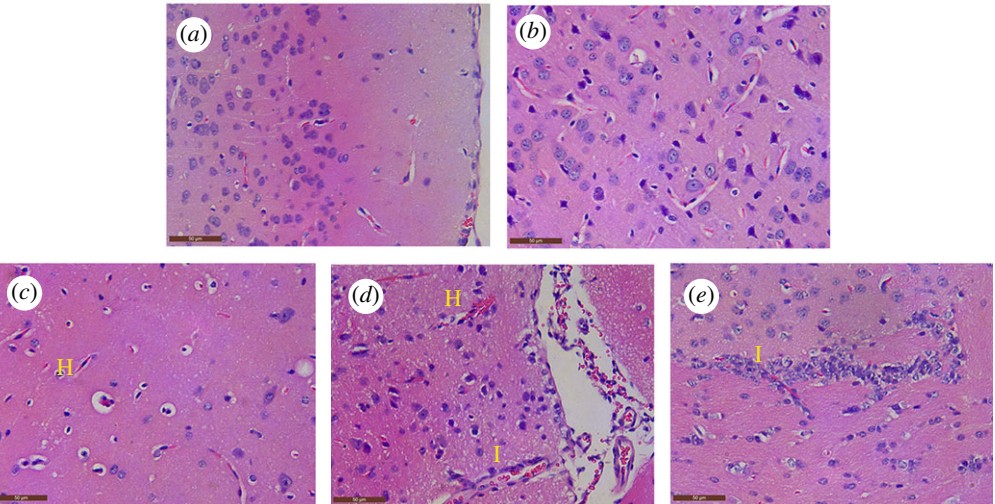

**Figure 1.** Histopathological observations of the brain of PRV-inoculated examined by H&E staining. (*a*) Microscopic lesion in the control group; (*b*) 36 hpi group; (*c*) hyperemia (H, yellow letter) appeared in the brain tissue, followed by perivascular space widened; perivascular lymphocytes increased, and degeneration and necrosis occurred in some neurons in the 48 hpi group; (*d,e*) focal inflammatory cellular infiltration (I, yellow letter) was found in the brain of the 60 hpi and 72 hpi groups.

brain structure was found in the control and PRV-inoculated mice before 36 hpi (figure 1*a,b*). At 48 hpi, hyperemia appeared in brain tissue, followed by perivascular space widening; perivascular lymphocytes increased, and degeneration and necrosis occurred in some neurons (figure 1*c*). Focal inflammatory cellular infiltration was found in the brain in 60–72 hpi (figure 1*d,e*).

## 3.2. The mRNA expression levels about innate immune-related genes

Innate immunity recognizes invading pathogens by binding to pattern recognition receptors, leading to the expression of antiviral molecules. Interferons (IFN) are antiviral molecules, which have a pivotal role in the clearance of invading pathogenic microorganisms [10]. In this work, the transcriptional levels of IFN-α, IFN-β and IFN-γ were measured by qRT-PCR.

As shown in figure 2, the expression levels of IFN-α and IFN-β in the brain of PRV infected mice were upregulated from 36 hpi ($p < 0.01$) and peaked at 48 hpi ($p < 0.01$) and then downregulated in 60–72 hpi ($p < 0.05$, figure 2*a,b*) . Furthermore, IFN-γ expression in the brain was downregulated before 36 hpi and then upregulated until 60 hpi ($p < 0.01$, figure 2*c*). In addition, pro-inflammatory cytokines (TNF-α, IL-1β, IL-6 and IL-18) and anti-inflammatory cytokines (IL-4 and IL-10) were also measured in the brain, respectively. The relative mRNA levels of TNF-α and IL-6 in the brain were remarkedly upregulated, caused by PRV infection, and peaked at 60 hpi ($p < 0.01$), and were then downregulated at 72 hpi (figure 2*d,f*). After PRV infection, the IL-1β expression in the brain was upregulated from 36 hpi and lasted for the whole experiment period ($p < 0.01$, figure 2*e*). In addition, IL-18 expression was upregulated from 36 hpi and had the same tendency as IL-1β ($p < 0.01$, figure 2*g*). The cytokine storms caused by IFN-α, IFN-β, TNF-α, IL-1β, IL-6 and IL-18 were related to the histopathological changes induced by PRV (figure 1). Remarkably, the mRNA expression levels of IL-4 and IL-10 were upregulated before 48 hpi ($p < 0.01$) and downregulated and maintained a low level until 72 hpi (figure 2*h,i*). This indicated that PRV inhibited the expression of IL-4 and IL-10 since 48 hpi.

## 3.3. Detection of pyroptosis cells in the brain

Pyroptosis in the brain was measured by Annexin V/PI double staining through flow cytometry. As shown in figure 3, a large number of PI + pyroptosis cells in the brain was observed at 36, 48, 60 and 72 hpi, which increased in a time-dependent way ($p < 0.01$) compared with that in the control group (electronic supplementary material, table S2). In addition, PRV significantly upregulated *in situ* the protein expression of Caspase-1 (figure 4*b–d*), IL-1β (figure 4*f–h*) and IL-18 (figure 4*j–l*) in the brain tissue in a time-dependent manner.

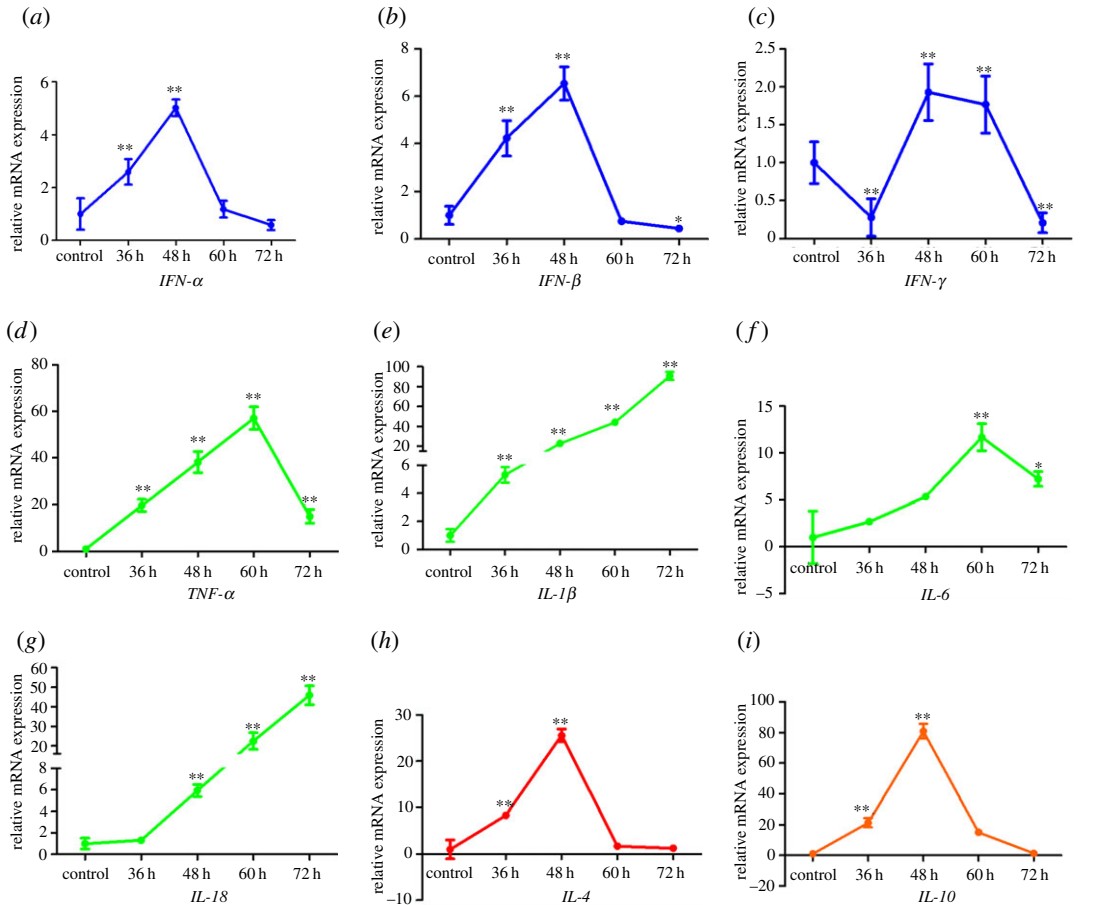

**Figure 2.** The dynamic changes of mRNA expression related to immune system induced by PRV (a) IFN-α, (b) IFN-β, (c) IFN-γ, (d) TNF-α, (e) IL-1β, (f) IL-6, (g) IL-18, (h) IL-4 and (i) IL-10. $^{**}p < 0.01$ and $^{*}p < 0.05$ compared with the control group.

## 3.4. Changes in protein expression levels related to pyroptosis in the brain

To further investigate the effect of PRV on the pyroptosis *in vivo*, the proteins involved in the pyroptosis signal pathway were detected by blotting, including NLRP3, Caspase-1, GSDMD, IL-1β and IL-18. From the result in figure 5, NRRP3 decreased at 36 hpi followed by an increasing tendency in a time-dependent manner. In addition, the Caspase-1 level was significantly upregulated by PRV from 36 hpi and maintained at a high level until the end of this experiment by comparing it with that in the control group. Furthermore, GSDMD, a pyroptosis marker protein, was also measured in all groups. As shown in figure 5a, the amount of GSDMD was higher in all PRV groups than in the control. In addition, the expression levels of IL-1β and IL-18, two cytokines related to pyroptosis, were determined by blotting. The result demonstrated that PRV could elevate the protein levels of IL-1β and IL-18.

## 3.5. Relative mRNA expression of genes related to pyroptosis in the brain

The mRNA expression levels of pyroptosis-related factors were further detected by qRT-PCR in this research. As shown in the electronic supplementary material, table S3, the mRNA expression level of *NRRP3* was markedly increased ($p < 0.05$) in a time-dependent manner following PRV treatment. Moreover, the mRNA expression levels of *Caspase-1* in the PRV inoculation groups were also upregulated ($p < 0.01$) compared to that among groups. Besides, the *GSDMD* mRNA expression level was upregulated in an increased tendency ($p < 0.01$) compared with that in the control group.

## 4. Discussion

PRV is a kind of neurophilic α herpesvirus, which belongs to the genus *Varicellovirus*, family *Herpesviridae*. Pigs are the only natural hosts of PRV for their survival to infection. However, mice and

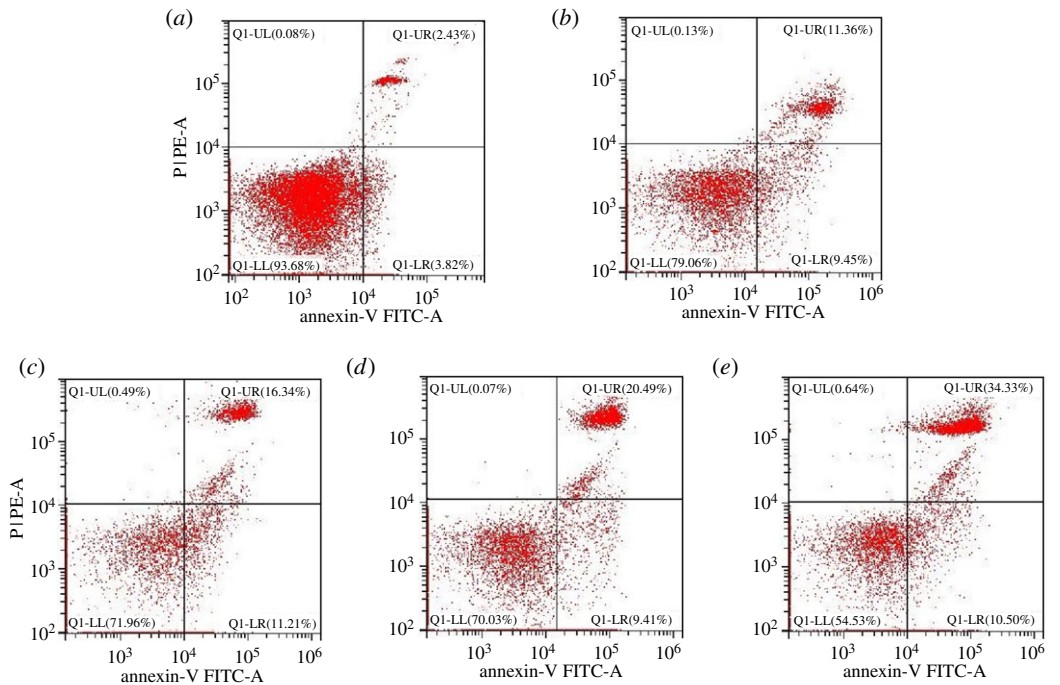

**Figure 3.** PRV caused pyroptosis in mice brains measured by the flow cytometry method through Annexin V-FITC/PI staining (*a*) control group, (*b*) the per cent of pyroptosis cells in 36 hpi group, (*c*) 48 hpi group, (*d*) 60 hpi group and (*e*) 72 hpi group.

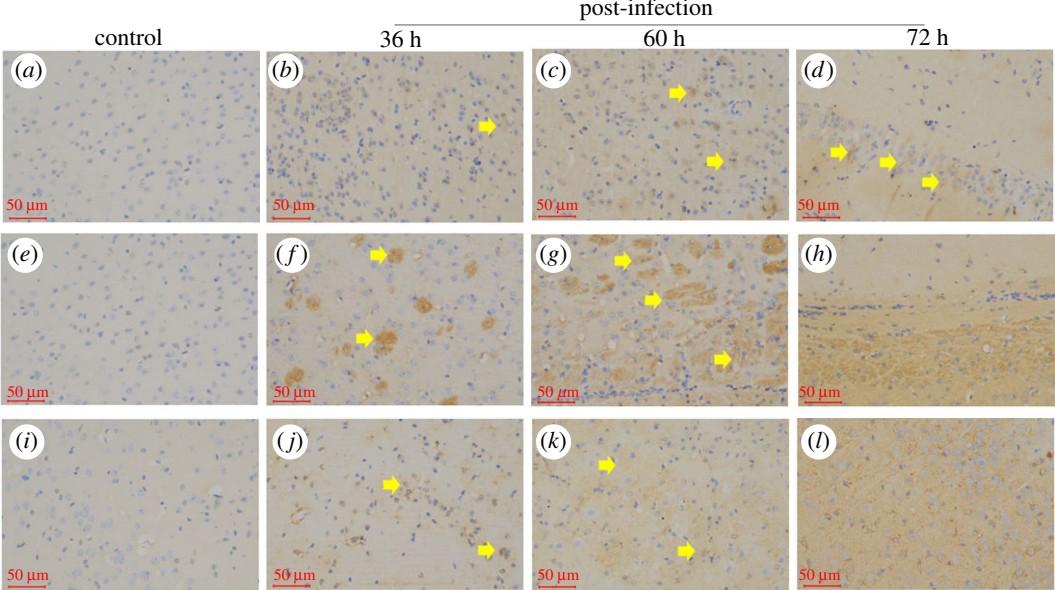

**Figure 4.** The *in situ* expression of Caspase-1, IL-1$\beta$ and IL-18 in the mice brains by HI. (*a*) Caspase-1 expression in the control group; (*b–d*) Caspase-1 expression in the brain at 36, 60 and 72 hpi; (*e*) IL-1$\beta$ expression in the control group; (*f–h*) IL-1$\beta$ expression in the brain at 36, 60 and 72 hpi; (*i*) IL-18 expression in the control group; (*j–l*) IL-18 expression in the brain at 36, 60 and 72 hpi; obvious positive regions could be seen in yellow in (*h*) and (*j*); positive regions are shown in yellow arrowheads in (*b–d,f,g,j* and *k*), scale bar = 50 μm.

rats can be naturally infected with PRV and cause a fatal disease. In the laboratory, after intranasal infection in adult mice, PRV enters peripheral nerve cells and spreads to the CNS [11]. Previous studies on PRV mainly focused on pathogenicity and the resulting host immune response. The purpose of this study was to research the kinetics of cytokine secretion *in vivo* and to clarify whether cytokine storm is involved in the pathogenesis of PRV. In this work, we identified cytokine storm and pyroptosis as the main causes of rapid death in mice infected with PRV.

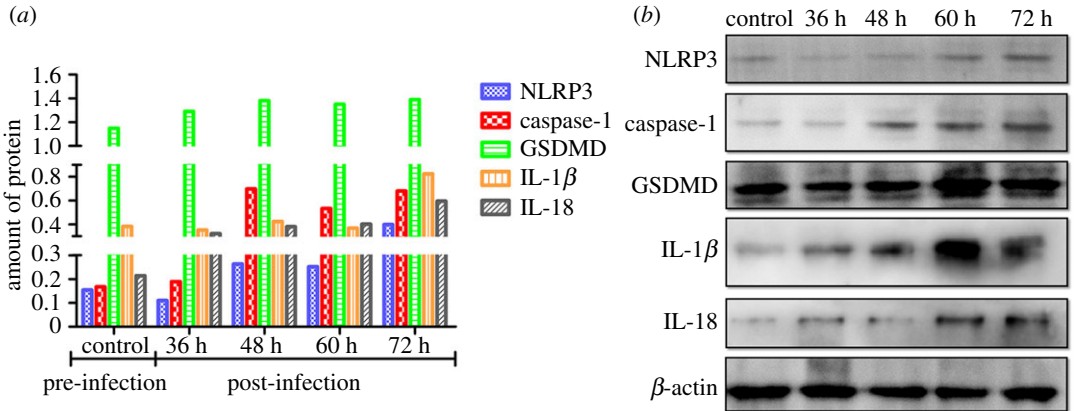

**Figure 5.** The protein expression levels were related to pyroptosis induced by PRV. (*a*) The relative expression amount protein of NRLR3, Caspase-1, GSDMD and IL-1$\beta$/18 to $\beta$-actin was present as a different colour and (*b*) protein expression related to pyroptosis measured by blotting.

The innate immune system is the first line of defense against the invasion of pathogens, accompanied by the recruitment of immune cells and inflammatory response [12]. Inflammation is a process to solve microbial infection and a complex process involving the regulation of cytokine production. Dysfunction of these factors can induce cytokine storms and related multiple organ failure [13]. An inflammatory response is usually caused by various pro-inflammatory cytokines, such as TNF, IL-1 and IL-6. These cytokines are the kinds of pleiotropic proteins involved in the regulation of cell death in inflammatory tissue, vascular endothelial cell permeability, attracting the blood cells to inflammatory tissue and acute-phase proteins production [14].

Cytokines play an important role in aspects of the immune response, coordinating the innate and adaptive immune response. Consequently, in most cases, cytokines play a protective role in resisting endogenous and exogenous noxious stimuli, such as tissue injury and microbial invasion. IFNs are recognized as the central factors of antiviral infection, which has a pivotal role in innate immune response [15]. In addition, cytokines play an important role in the pathogenesis of antiviral and viral infections. However, excessive immune activation and excessive release of cytokines could be rather pernicious [16]. For example, over-expression of TNF-α, IL-1 and IL-6 in the immune system could lead to vascular leakage, systemic fatigue, cardiomyopathy and acute-phase protein synthesis [16]. In addition, persistent excessive IFN-α/β may also be harmful to the immune system [15]. In the present study, we found that cytokine storm was induced by PRV in mice brains from 36 to 72 hpi, including the elevated expression levels of Type I IFNs (IFN-α and IFN-β) and Type II IFNs (IFN-γ) as well as pro-inflammatory factors compared to that in the control group ($p < 0.01$). This storm was consistent with the histopathological changes, including hyperemia and inflammatory cell infiltration in mice brains. Studies have shown that PRV could regulate the expression of cytokines, including Type I and Type II IFN and inflammatory factors, to establish a successful infection [17,18]. Furthermore, IFN-α and IFN-β mediate a positive feedback regulation by binding to IFN-α and IFN-β receptor in an automatic or paracrine manner [19].

PRV infection can induce apoptosis, which has been reported previously *in vitro* and *in vivo* [20,21]. However, apoptosis is usually considered as an insoluble programmed cell death (PCD), which is characterized by an active programmed process of cell decomposition to avoid inflammation [22]. The discrepancy found in this research may be induced by a new kind of PCD in the cell death process. Pyroptosis is a new type of pro-inflammatory cell death, which is emerging as the mechanism of antagonizing and clearing pathogen infection and requires the activation of Caspase1/4/5/11 [23]. In this research, an increasing tendency of pyroptotic cells induced by PRV was detected in the brain through flow cytometry by comparing with the control group ($p < 0.01$). Furthermore, the expression of genes and proteins related to the pyroptosis pathway were elevated by qRT-PCR and blotting methods, respectively. Inflammasomes are cytosolic sensors that could activate Caspase-1 [24]. Once activated, Caspase-1 can process and maturates IL-1β and IL-18 precursors, as well as cleaving GSDMD, resulting in cell membrane channel opening and the pyroptosis [9]. Among the inflammasomes, NLRP3 is currently the most well-known one, which responds to various stimuli. NLRP3 was activated by PRV in this research. In addition, GSDMD, a key executor in pyrotosis [25],

was also activated by PRV. This activity of GSDMD leads to the indirect release of IL-1$\beta$ and IL-18 from membrane pores [25]. The pyroptotic cell-fate decision provides a large amount of inflammatory response at the site of infection. This was consistent with the results from histopathological analysis (figure 1) and immunohistochemistry (figure 4), as well as a full explanation for cytokine storm caused by PRV in mice brains. However, cytokine storms and pyroptosis might cause the rapid death of mice caused by PRV strain. Furthermore, due to the difference in immune systems between mice and pigs, more detailed information about the pathogenesis to pigs and other mammals needs to be further clarified in the future.

# 5. Conclusion

Cytokine storms and pyroptosis might be the cause of rapid death in mice inoculated with PRV strain. These results provided a new insight for further understanding the pathogenesis caused by PRV.

Ethics. This study was approved by Animal Care and Use Committee of Tongren Polytechnic College and Dossy Experimental Animal Corporation (SYXK2014-189). All animal administrations, sample collection and procedures were performed according to the approved guidelines.

Data accessibility. The datasets supporting this article have been uploaded in the form of 'electronic supplementary material'. In addition, the raw material related to this work are deposited at Dryad (https://doi.org/10.5061/dryad.zw3r2287j) [26]. The data are provided in the electronic supplementary material [27].

Authors' contributions. W.S. conceived this study, revised the manuscript, participated in the data analysis and artwork making; S.L. did the qRT-PCR and Western blotting experiment and participated in the design of this study; R.Y. did the animal experiment. X.H. and J.Y. carried out a final revision of the manuscript.

Competing interests. The authors declare that there is no conflict of interest to this research.

Funding. This research was supported by Guizhou Science and Technology Planning Project (grant no. [2019]1456) and Tongren Science and Technology Planning Project (grant no. [2019]12-6).

Acknowledgements. Many thanks to Jiangtao Feng for his help in animal administration.

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
