## [Peer Review File · Royal Society Open Science]

Review History

RSOS-210296.R0 (Original submission)

Review form: Reviewer 1

Is the manuscript scientifically sound in its present form?

No

Are the interpretations and conclusions justified by the results?

Yes

Is the language acceptable?

No

Do you have any ethical concerns with this paper?

No

Have you any concerns about statistical analyses in this paper?

Yes

Recommendation?

Major revision is needed (please make suggestions in comments)

Comments to the Author(s)

This paper has some merit addressing an important issue about the role of the immune response in the pathogenesis of the ADV infection in the mouse model. However, it needs a profound revision on the language, more detail in the figures texts, and a complete description of the experimental design (groups and number of animals). Also I would recommend to discuss the evident limitations of the mouse model when extrapolating to economical species suffering of Pseudorabies. The statistical results also should be more explicit in the text and figures to strengthen the discussion. See Appendix A.

Decision letter (RSOS-210296.R0)

Dear Dr Sun

The Editors assigned to your paper RSOS-210296 "Cytokine storms and pyroptosis are primarily responsible for the rapid death of mice infected with pseudorabies virus" have now received comments from reviewers and would like you to revise the paper in accordance with the reviewer comments and any comments from the Editors. Please note this decision does not guarantee eventual acceptance.

We invite you to respond to the comments supplied below and revise your manuscript. Below the referees' and Editors' comments (where applicable) we provide additional requirements. Final acceptance of your manuscript is dependent on these requirements being met. We provide guidance below to help you prepare your revision. There are particular concerns regarding the quality of the written English - please seek advice from a service such as <https://royalsociety.org/journals/authors/benefits/language-editing/> before resubmitting.

Please submit your revised manuscript and required files (see below) no later than 21 days from today's (ie 10-Jul-2021) date. Note: the ScholarOne system will 'lock' if submission of the revision is attempted 21 or more days after the deadline. If you do not think you will be able to meet this deadline please contact the editorial office immediately.

on behalf of Prof Malcolm White (Subject Editor)
openscience@royalsociety.org

Reviewer comments to Author:

Reviewer: 1

Comments to the Author(s)

This paper has some merit addressing an important issue about the role of the immune response in the pathogenesis of the ADV infection in the mouse model. However, it needs a profound revision on the language, more detail in the figure legends, and a complete description of the experimental design (groups and number of animals). Also I would recommend to discuss the evident limitations of the mouse model when extrapolating to economical species suffering of Pseudorabies. The statistical results also should be more explicit in the text and figures to strengthen the discussion.

Going through the experimental design, it is not clear how the groups of animals were conformed. A total of eighty mice were divided into two groups (probably 40 control and 40 experimental), however, 6 animals of each group were killed at 4 different time points, given a total of 24 mice per group, ¿what happened with the rest of the animals?

On figure 2 six time points were described (36 mice), however, no results on the control group were shown.

All figures need a self-explanatory text. Figure 2 misses IL-18 text.

Although a statistical analysis is claimed to be applied on the results, differences were not shown in the text or on the figures. Since the groups of animals were not accurately described, the analysis must be detailed.

===PREPARING YOUR MANUSCRIPT===

While not essential, it will speed up the preparation of your manuscript proof if accepted if you format your references/bibliography in Vancouver style (please see

<https://royalsociety.org/journals/authors/author-guidelines/#formatting>). You should include DOIs for as many of the references as possible.

===PREPARING YOUR REVISION IN SCHOLARONE===

Author's Response to Decision Letter for (RSOS-210296.R0)

See Appendix B.

RSOS-210296.R1 (Revision)

Review form: Reviewer 1

Is the manuscript scientifically sound in its present form?

Yes

Are the interpretations and conclusions justified by the results?

Yes

Is the language acceptable?

No

Do you have any ethical concerns with this paper?

No

Have you any concerns about statistical analyses in this paper?

No

Recommendation?

Accept with minor revision (please list in comments)

Comments to the Author(s)

This paper is now suitable for publication. However, requires a further review of the language to make it more understandable. Please check up the comments made on the yellow marked words or phrases, as well as on the stoke out words in the reviewed PDF file and be sure to amend the errors.

Decision letter (RSOS-210296.R1)

Dear Mr Sun

On behalf of the Editors, we are pleased to inform you that your Manuscript RSOS-210296.R1 "Cytokine storms and pyroptosis are primarily responsible for the rapid death of mice infected with pseudorabies virus" has been accepted for publication in Royal Society Open Science subject to minor revision in accordance with the referees' reports. Please find the referees' comments along with any feedback from the Editors below my signature.

Please submit your revised manuscript and required files (see below) no later than 7 days from today's (ie 02-Aug-2021) date. Note: the ScholarOne system will 'lock' if submission of the revision is attempted 7 or more days after the deadline. If you do not think you will be able to meet this deadline please contact the editorial office immediately.

on behalf of Professor Malcolm White (Subject Editor)
openscience@royalsociety.org

Associate Editor Comments to Author:

Please ensure that you carefully revise the paper to take into account the remaining queries/comments of the reviewer. As you have been requested to edit the written English, you must provide proof that you have done so: acceptable proof includes a certificate of language-editing from a language editing service or a signed letter from a native speaker of English. If you do not provide this proof, your manuscript may be returned to you.

For information about language editing services endorsed by the Royal Society, please follow the link below:
<https://royalsociety.org/journals/authors/language-polishing/>

Reviewer comments to Author:
Reviewer: 1
Comments to the Author(s)

This paper is now suitable for publication. However, requires a further review of the language to make it more understandable. Please check up the comments made on the yellow marked words or phrases, as well as on the stoke out words in the reviewed PDF file and be sure to amend the errors.

===PREPARING YOUR MANUSCRIPT===

===PREPARING YOUR REVISION IN SCHOLARONE===

Please ensure that you include a summary of your paper at Step 2 'Type, Title, & Abstract'. This should be no more than 100 words to explain to a non-scientific audience the key findings of your

research. This will be included in a weekly highlights email circulated by the Royal Society press office to national UK, international, and scientific news outlets to promote your work.

Author's Response to Decision Letter for (RSOS-210296.R1)

See Appendix C.

Decision letter (RSOS-210296.R2)

Dear Mr Sun,

I am pleased to inform you that your manuscript entitled "Cytokine storms and pyroptosis are primarily responsible for the rapid death of mice infected with pseudorabies virus" is now accepted for publication in Royal Society Open Science.

on behalf of Malcolm White (Subject Editor)
openscience@royalsociety.org

Appendix A**ROYAL SOCIETY
OPEN SCIENCE****Cytokine **stroms** and pyroptosis are primarily responsible
for the rapid death of mice infected with pseudorabies virus**

Journal:	Royal Society Open Science
Manuscript ID	RSOS-210296
Article Type:	Research
Date Submitted by the Author:	24-Feb-2021
Complete List of Authors:	Liu, Shanshan; Tongren Polytechnic College; National and Local Engineering Research Centre for Separation and Purification Ethnic Chinese Veterinary Herbs Sun, Wei; Tongren Polytechnic College Huang, Xuefei; Tongren Polytechnic College Yuan, Rui; Tongren Polytechnic College Yu, Jiansheng; Tongren Polytechnic College; National and Local Engineering Research Centre for Separation and Purification Ethnic Chinese Veterinary Herbs
Subject:	biotechnology < BIOLOGY, health and disease and epidemiology < BIOLOGY
Keywords:	cytokine stroms, pyroptosis, pseudorabies virus
Subject Category:	Biochemistry, Cellular and Molecular Biology

Author-supplied statements

Relevant information will appear here if provided.

Ethics

Does your article include research that required ethical approval or permits?:

Yes

Statement (if applicable):

The authors declare that this manuscript is original, which has not been published before and is also not currently being considered for publication elsewhere.

All animal procedures were approved by the Tongren Polytechnic College Animal Care and Use Committee and were in accordance with the NIH Guide for the Care and Use of Laboratory Animals.

Data

It is a condition of publication that data, code and materials supporting your paper are made publicly available. Does your paper present new data?:

Yes

Statement (if applicable):

The datasets supporting this article have been uploaded in the form of "Supplementary Material". In addition, the material related to this **wirk** are deposited at Dryad (https://datadryad.org/stash/share/2URdlvjkcYyMv4hp7II9NWoj-_gwX3Vsx4uZ-h_Gmk)

Conflict of interest

I/We declare we have no competing interests

Statement (if applicable):

CUST_STATE_CONFLICT :No data available.

Authors' contributions

This paper has multiple authors and our individual contributions were as below

Statement (if applicable):

Shanshan Liu did the molecular lab work and drafted the manuscript; Wei Sun conceived this study and participated in the data analysis and artwork making; Rui Yuan did the animal experiment.

Xuefei Huang and Jiansheng Yu gave a finally revised the manuscript.

Cytokine storms and pyroptosis are primarily responsible for the rapid death of mice infected with pseudorabies virus

Shanshan Liu^{1,2}, Wei Sun^{1*}, Xuefei Huang¹, Rui Yuan^{1,2} and Jiansheng Yu^{1,2}

¹College of Agriculture, Tongren Polytechnic College, Bijiang District, Tongren City, Guizhou, 554300, China

²National and Local Engineering Research Centre for Separation and Purification Ethnic Chinese Veterinary Herbs, Tongren City, Guizhou, 554300, China

Keywords: cytokine storms; pyroptosis; pseudorabies virus

Abstract

Pseudorabies virus (PRV), the causative agent of Aujeszky's disease (AD), is one of the most harmful pathogens to pig industry. PRV has the ability to infect and kill a variety of mammals. Nevertheless, the underlying pathogenesis related to PRV is still unclear. This study aims to investigate the pathogenesis induced by PRV in a mouse model. The mice infected with PRV-HLJ strain developed severe clinical manifestations at 36 hours post infection (hpi), and mortality occurred within 48–72 hpi. Hematoxylin eosin staining and qRT-PCR methods were used to detect the pathological damage and expression of cytokines related to immune in brain tissue, respectively. The cytokine storms caused by IFN- α , IFN- β , TNF- α , IL-1 β , IL-6 and IL-18 was related to the histopathological changes induced by PRV. This pattern of cytokine secretion clearly depicts an image of a typical cytokine storm, characterized by dysregulated secretion of pro-inflammatory cytokines and imbalanced pro-inflammatory and anti-inflammatory responses. In addition, the pyroptosis pathway was also activated by PRV through elevating the expression levels of nod-like receptor protein 3, caspase-1, gasdermin-D and interleukin-1 β /18. These findings provide a way for further understanding of the molecular basis in PRV pathogenesis.

1. Introduction

Pseudorabies virus (PRV) or swine α herpesvirus 1 is the pathogen of porcine Aujeszky disease (AD), which causes symptoms of respiratory system, nervous system and reproductive system¹. Besides pigs, many mammals are susceptible to PRV infection, such as cattle, sheep, rabbits, cats, dogs, guinea pigs, rats and mice². However, pigs are the only susceptible animals that can survive, although the prognosis of the disease largely depends on the factors including inoculation site, virus strain and titer as well as age of pigs *et. al.*³. AD is a highly infectious disease with high mortality in piglets. Transmission mainly occurs through direct contact with oral and nasal secretions, but can also occur through aerosol through placenta or sexual intercourse¹. Therefore, the prevalence of PRV has led to a wide range of economic losses in the pork production industry. Inactivated and attenuated vaccines have been developed to delay or reduce swine death, but they are not yet able to eradicate the disease because none of them prevent potential infection and reactivation and shedding of virulence field virus⁴. Due to the serious impact of AD on pig industry, some countries are trying to eradicate AD basing on the DIVA (Differentiating Infected From Vaccinated animals) program. However, since 2011, the outbreak of AD occurred in farm pigs vaccinated with PR vaccine in China, which indicates that AD vaccine can not provide effective treatment to prevent wild PR infection⁵.

Mice and rabbits are usually used to study PRV in the laboratory. After infection, the animals showed abnormal excitement and nasal itching, accompanied by convulsions, and rapid death. In mice, PRV almost completely manifested as a neurogenic infection of the central nervous system (CNS), accompanied by fulminant central nervous symptoms and high mortality⁶. PRV is known to cause severe encephalitis in piglets, various non-native hosts, even in humans⁷. Few studies have focused on the pathogenesis of encephalitis. It is known that pyroptosis involved in the immune response in various types of cell, which can be triggered by a variety of pathological stimuli, leading to the secretion of proinflammatory cytokines and intracellular contents⁸. Inflammation is a double-edged sword, which has a crucial role in metabolism. Mild inflammatory response could protect the body to a certain degree,

*Author for correspondence (sunwei_223@163.com).

†Present address: College of Agriculture, Tongren Polytechnic College, Bijiang District, Tongren City, Guizhou, 554300, China

50 help to repair damaged tissue, and be beneficial to steady-state reconstruction. Nevertheless, excessive inflammation
51 may form a "cytokine storm", leading to tissue damage. In the present work, we describe the influence of PRV on the
52 immune factor and pyroptosis-related factor in mice brain.

2. Materials and Methods

2.1 Reagents and animals

55 Annexin V-FITC/PI Apoptosis Kit was obtained from B D Company (Franklin Lakes, USA). Modified Bradford
56 Protein Assay Kit, Antibody against NLRP3, Animal Total RNA Isolation Kit and Tissue Total Protein Extraction
57 Kit were supplied by Sagon Biotech Company (Shanghai, China). Another antibody against Gasdermin D (GSDMD)
58 was bought from Thermo Fisher Company (USA). IL-1 β , IL-18 and β -actin antibodies were obtained from Bioss
59 Company (Beijing, China). Caspase-1 antibody was obtained from Boster Biological Technology com.Ltd (Wuhan,
60 Hubei, China). PrimeScript RT reagent Kit was bought from TAKARA company (Daliang, China). PRV-HLJ strain
61 (MK080279.1) isolated Heilong Jiang was provided by Professor Jingfei Wang from Harbin Veterinary Research
62 Institute, CAAS. Eighty 6-week-old female Balb/C mice were obtained from Dossy Experimental Animal
63 Corporation (Chengdu, China).

2.2 Experimental

65 One week later, the mice were divided randomly into two groups, i.e control and post-infection groups. The mice in
66 the first group were injected via hypodermic injection with 0.2 mL of physiological normal saline (NS). The mice in
67 the second group was received with 0.2 mL PRV-HLJ strain(10^4 TCID₅₀). The animals were fed in the room
68 illuminated with a 12 h light-dark cycle. The ambient temperature and relative humidity maintained at 22-24°C and
69 40-60%, respectively. Ten mice were fed in each cage and given the above dosage. Water and diet was provided ad
70 libitum. Six mice from each group were killed at 36, 48, 60 and 72 hour post infection (hpi), and their brain tissues
71 were collected aseptically. Subsection specimens were snap frozen and stored at -80°C for RNA extraction. In
72 addition, portions of the brain were fixed in 4% paraformaldehyde solution for histopathological examination.

2.3 Histopathological analysis

74 Histopathological observation was operated by using a standard laboratory procedure. The brain was removed from
75 experimental animals and washed thoroughly in phosphate buffered saline (PBS, pH 7.4). Then, the tissue was fixed
76 in 4% paraformaldehyde for 2 d and transferred to 95% percent or absolute alcohol for dehydration. Thereafter,
77 processed to paraffin embedding routine. The paraffin-embedded tissue was sliced into a 5 μ m section, dewaxed in
78 xylene and then rehydrated in graded alcohols. Section was stained with hematoxylin and eosin (H.E) staining and
79 then examined under light microscope for histopathological examination.

2.4 Detection of pyroptosis by flow cytometry in brain

81 The rate of pyroptosis cells in brain was measured using an annexin V-FITC/PI Apoptosis kit according to the
82 manuscript's protocol. Brains were taken from mice, which were humanely killed at time above mentioned, ground to
83 form suspension and filtered with a 300-mesh nylon screen. The cells were washed three times with pre-cold PBS
84 and adjusted at a concentration of 1×10^6 cells/mL. Furthermore, 100 μ L cells were incubated with annexin V-
85 FITC/PI staining at room temperature for 15 min in a culture tube in the dark atmosphere. Each tube was added with
86 300 μ L of binding buffer and then detected with a FCM (Becton Dickinson, USA). CellQuest Pro software (Becton
87 Dickinson, USA) was used to visualize the results.

2.5 RNA extraction and qRT-PCR analysis

90 Total RNA from brains were isolated by a "Animal Total RNA Isolation Kit" according to the kit directions. RNA
91 integrity was detected by 2% agarose gel electrophoresis. A spectrophotometer was used to detect the RNA quantity
92 and quality (NanoDrop-2000, ThermoFisher Company, USA). Total RNA was converted into DNA for qRT-PCR by
93 using the PrimeScript RT reagent Kit. The first strand of cDNA was amplified through SYBR staining on a
94 LightCycler 96 apparatus (Roche, Germany). All primers used in this research were designed by Oligo 7 software
95 and synthesized by Sagon Biotech Ltd., (Shanghai, China). Detailed information about primers were available in
96 Table S1. 2^{- $\Delta\Delta$ Ct} method was used for the analysis of mRNA expression. β -actin, a housekeeping gene, was used as
97 an internal control for values correction.

2.6 Western blotting

100 A "Tissue Total Protein Extraction Kit" was used for protein extracted from brain. Protein concentration of each
101 specimen was measured by Bradford assay. The proteins were first separated on a 10% sodium dodecyl sulfate
102 polyacrylamide gel electrophoresis (SDS-PAGE) followed by being transferred onto a polyvinylidene difluoride
103 (PVDF) membrane. Then, the PVDF membrane was blocked with TBST solution contained with 5% non-fat milk at

room condition for 2 h, and then incubated at 4°C condition for 12 h with the corresponding primary antibodies diluted with a solution: NLRP3(1:1000), Caspase-1(1:800), GSDMD(1:900), IL-1β(1:500), IL-1β(1:500) and β-actin(1:2000). Then, the PVDF membrane was incubated at room temperature with HRP-labeled secondary antibody for 1 h and blots were measured by a ECL reagent. The β-actin was used as a protein loading control.

2.7 Statistical analysis

SPSS22.0 software was used to data analyses. The results were espresented as means ± standards deviations(mean±SD). One-way analysis of variance (ANOVA) package in SPSS 22.0 was used to evaluate the statistical significances between PRV and control group. GraphPad Prism6.0 software was used to perform statistical artworks. In all statistical comparisons, *p* value was introduced as a judgment to the statistical difference.

3. Results

3.1 Clinical symptoms and histopathological analysis

At 36 hpi, the mice infected with PRV generally showed typical clinical signs, including depression, anorexia and neuropathic itch. Mortality occurred within the period of 48-72 hpi. However, none of the mice in control group exhibited clinical symptoms or died.

Compared with the control group, there was a significant difference in the microscopic lesions in the PRV-inoculated mice (Figure 1). Normal brain structure was found in the control and PRV-inoculated mice before 36 hpi (Figure 1A and B). At 48 hpi, hyperemia appeared in brain tissue follow by perivascular space widened, perivascular lymphocytes increased, as well as degeneration and necrosis occurred in some neurons (Figure 1C). Focal lymphocytic infiltration was found in the brain at 60-72 hpi (Figure D and E).

3.2 The mRNA expression levels about innate immune-related genes

Innate immunity recognizes invading pathogens by binding to pattern recognition receptors, leading to the expression of antiviral molecules. Interferons (IFN) is an antiviral molecule, which has a pivotal role in the clearance of invading pathogenic microorganisms⁹. In this work, the transcriptional levels of IFN-α, IFN-β and IFN-γ were determined by qRT-PCR.

As shown in Figure 2, the expression levels of IFN-α and IFN-β in the brain of PRV infected mice were up-regulated at 24 hpi and peaked at 48 hpi (Figure 2A-B). Interesting, IFN-γ expression in brain was down-regulated before 36 hpi and then up-regulated until 60 hpi (Figure 2C). In addition, pro-inflammatory cytokines (TNF-α, IL-1β, IL-6 and IL-18) and anti-inflammatory cytokines (IL-4, IL-10) were also measured in the brain, respectively. The relative mRNA levels of TNF-α and IL-6 in brain were remarkably up-regulated caused by PRV infection, and peaked at 60 hpi followed by down-regulated but maintain at a high levels until 72 hpi (Figure 2D and F). After PRV infection, the IL-1β expression in brain was up-regulated from 24 hpi and lasted for the whole experiment period. However, IL-1β expression was up-regulated from 36 hpi and had a same trend as IL-1β. The cytokine stroms caused by IFN-α, IFN-β, TNF-α, IL-1β, IL-6 and IL-18 was related to the histopathological changes induced by PRV (Figure 1). Remarkably, the mRNA expression levels of IL-4 and IL-10 was up-reguated before 48 hpi and rapid down-regulated and maintained a low level until 72 hpi (Figure 2H and I). This indicated that PRV inhibited the expression of IL-4 and IL-10 since 48 hpi.

3.3 Detection of pyrolic cells in brain

Pyroptosis in brain was measured by Annexin V/PI double staining through flow cytometry. According to Figure 3, a large number of of PI+ pyroptotic cells in brain observed at 36, 48, 60 and 72 hpi, which increased remarkably in a time-dependent way (*p*<0.01) compared with than in the control group (Table S2). In addition, PRV significantly up-regulated *in situ* the protein expression of caspase-1 (Figure 4B-D), IL-1β (Figure 4F-H) and IL-18 (Figure 4J-L) in the brain tissue in a time-dependent manner.

3.4 Changes of protein expression levels related to pyroptosis in the brain

To further investigate the effect of PRV on pyroptosis *in vivo*, the proteins involved in pyroptosis signal pathway were detected by western blotting, including NLRP3, Caspae-1, GSDMD, IL-1β and IL-18. From the result in figure 5, NRRP3 decreased at 36 hpi followed by an increasing trendency in a time-dependent manner. In addition, the Caspase-1 level was significantly up-regulated by PRV from 24 hpi and manitained at a high level until the end of this experiment by compareing with that in the control group. Furthermore, GSDMD, an important pyroptosis marker protein, was also measured in all groups. As shown in Figure 5A, the amount of GSDMD was higher in all PRV group compared to the control. In addition, the expression levlels of IL-1β and IL-18, two cytokines related to pyroptosis, were determined by western blotting. The result demonstrated that PRV could elevated the protein levels of IL-1β and IL-18.

3.5 Relative mRNA expression of genes related to pyroptosis in the brain

158 The mRNA expression levels of pyroptosis-related factors were further detected by qRT-PCR in this research. As
 159 shown in Table S3, the mRNA expression level of *NRRP3* were markedly increased ($p < 0.05$) in a time-dependent
 160 manner following by PRV treatment. Moreover, the mRNA expression levels of *Caspase-1* in the PRV inoculation
 161 groups were also up-regulated ($p < 0.01$) when by comparint that among groups. Besides, the *GSDMD* mRNA
 162 expression level was up-regulated in an increased tendency ($p < 0.01$) compared with that in the pre-infection group.

4. Discussion

164 PRV is a kind of neurophilic α herpesvirus, which belong to the genus *Varicelloviru*, family *Herpesviridae*. Pigs are
 165 the only natural hosts of PRV for their survive to infection. However, mice and rats can be naturally infected with
 166 PRV and cause a fatal disease. In lab, after intranasal infection in adult mice, PRV enters peripheral nerve cells and
 167 spreads to the central nervous system¹⁰. Previous studies on PRV mainly focused on the pathogenicity and the
 168 resulting host immune response. The aim of our research was to describe the kinetics of cytokine secretion *in vivo* and
 169 to clarify whether cytokine storm involved in the pathogenesis of PRV. In this work, we identified cytokine strom and
 170 pyroptosis as the main causes of rapid death in mice infected with PRV.

171 Innate immune system is the first line in defending against the invasion of pathogens, accompanied by the
 172 recruitment of immune cells and the initiation of inflammatory response¹¹. Inflammation is an important process to
 173 solve microbial infection and a complex process involving the regulation of cytokine production. Dysfunction of these
 174 mechanisms can induce cytokine storms and related multiple organ failure¹². Inflammatory response is usually caused
 175 by a variety of pro-inflammatory cytokines, such as TNF, IL-1 and IL-6. These cytokines are the kind of pleiotropic
 176 proteins, which involves in the regulation of cell death in inflammatory tissue, vascular endothelial cell permeability,
 177 attracting the blood cells to inflammatory tissue, and acute phase proteins production¹³.

178 Cytokines play an important role in all items of immune response, coordinating the innate and adaptive immune
 179 response. Consequently, in most cases, cytokines play a protective role in resisting endogenous and exogenous
 180 noxious stimuli, such as tissue injury and microbial invasion. IFNs are recognized as the central factors of antiviral
 181 infection, which has a pivotal role in innate immune response¹⁴. In addition, Cytokines and interleukin play an
 182 important role in the pathogenesis of antiviral and viral infection. However, excessive immune activation and
 183 excessive release of cytokines could be rather pernicious¹⁵. For example, overexpression of TNF- α , IL-1, and IL-6 in
 184 immune system could lead to vascular leakage, systemic fatigue, cardiomyopathy, vascular leakage, and acute phase
 185 protein synthesis¹⁵. In addition, persistent excessive IFN- α/β may also be harmful to immune system¹⁴. In the present
 186 study, we found that a strong cytokine strom was induced by PRV in mice brain from 36-72 hpi, including the
 187 elevated expression levles of Type I IFNs (IFN- α and IFN- β) and Type II IFNs (IFN- γ) as well as proinflammatory
 188 factors. This strom was consist with the histopathological changes in mice brain. Studies have shown that PRV
 189 could regulate the expression of cytokines, including type I and type II interferon and inflammatory factors, to
 190 establish a successful infection^{16, 17}. Furthermore, IFN- α and IFN- β mediates a positive feedback regulation by
 191 binding to IFN- α and IFN- β receptor in an automatic or paracrine manner¹⁸.

192 PRV infection causes apoptosis has been reproted previously *in vitro* and *in vivo*^{19, 20}. However, apoptosis is usually
 193 considered as an insoluble programmed cell death (PCD), which is characterized by an active programmed process of
 194 cell decomposition to avoid inflammation²¹. The discrepancy found in this research may be induced by a new kind of
 195 PCD in cell death process. Pyroptosis is a new type of pro-inflammatory cell death, which is emerging as the
 196 mechanism of antagonizing and clearing pathogen infection, and requires the activation of caspase 1/4/5/11²². In this
 197 research, the increasing tendency of pyroptotic cells induced by PRV were detected in brain through flow cytometry.
 198 Furthermore, the expression of genes and proteins related to pyroptosis pathway were elevated by qRT-PCR and
 199 Western Blotting methods, respectively. Inflammasomes are cytosolic sensors that could activate Caspase-1²³. Once
 200 activated, Caspase-1 has the ability to process and maturates IL-1 β and IL-18 precursors, as well as cleave GSDMD,
 201 resulting in cell membrane channel opening and pyroptosis⁸. Among the inflammasomes, NLRP3 is currently the
 202 most well-known one, which responds to a variety of stimuli. NLRP3 was accivated by PRV in this research. In
 203 addition, GSDMD, a key executor in pyrotosis²⁴, also be activated by PRV. This activity of GSDMD leads to the
 204 indirect release IL-1 β and IL-18 from membrane pores²⁵. The pyroptotic cell-fate decision provides a large amount of
 205 inflammatory response at the site of infection. This was consist with the results from histopathological analysis
 206 (Figure 1) and immunohistochemistry (Figure 4), as well as a full explanation for cytokine strom caused by PRV in
 207 mice brain.

5. Conclusion

208 Cytokine stroms and pyroptosis might be the main cause for the rapid death of mice inoculated with PRV strain. This
 209 results provided a new insight for further understanding the pathogenesis caused by PRV.

Acknowledgments

212 Many thanks to Jiangtao Feng for his helping in animal administration.

Ethical Statement

215 This study was approved by Animal Care and Use Committee of Tongren Polytechnic College. All animal administrations, sample
 216 collection and procedures were performed by the guidelines approved.

Funding Statement

219 This research was supported by Key Research Project of Guizhou Science and technology (No. [2019]1456), Tongren Science and
 220 Technology Project (No. [2019]12-6).

222
1 223
2 224
3 225
4 226
5 227
6 228
7 229
8 230
9 231
10 232
11 233
12 234
13 235
14 236
15
16
17
18
19
20
21
22
23
24
25
26
27
28
29
30
31
32
33
34
35
36
37
38
39
40
41
42
43
44
45
46
47
48
49
50
51
52
53
54
55
56
57
58
59
60

Data Accessibility

The datasets supporting this article have been uploaded in the form of “Supplementary Material”. In addition, the raw material related to this **wirk** are deposited at Dryad (<https://datadryad.org/stash/dataset/doi:10.5061/dryad.zw3r2287j>)²⁵.

Competing Interests

The authors declare that there is no conflict of interests to this research.

Authors' Contributions

Shanshan Liu did the qRT-PCR and Western blotting experiment and drafted the manuscript; Wei Sun conceived this study and participated in the data analysis and artwork making; Rui Yuan did the animal experiment. Xuefei Huang and Jiansheng Yu ~~gave a~~ finally revised the manuscript.

References

1. Alexander SV, Philipp GD, Marion S, Joachim SH, Anton SEH, Max S, Matthias K, Boris BL, Von BMS. 2018. Cytokine release syndrome. *J Immunother Cancer* 6, 56. (doi:10.1186/s40425-018-0343-9).
2. Broz P, Dixit VM 2016. Inflammasomes: mechanism of assembly, regulation and signalling. *Nat Rev Immunol* 16, 407-420. (doi: 10.1038/nri.2016.58)
3. Cheung AK, Chen Z, Sun Z, Mccullough D. 2000. Pseudorabies virus induces apoptosis in tissue culture cells. *Arch Virol* 145, 2193-2200. (doi: 10.1007/s007050070049)
3. Daniela, Tudor, and, Sabine, Riffault, and, Charles, Carrat, and, François. 2001. Type I IFN Modulates the Immune Response Induced by DNA Vaccination to Pseudorabies Virus Glycoprotein C. *Virology* 286, 197-205. (doi: 10.1006/viro.2001.0957)
4. Delva JL, Nauwynck HJ, Mettenleiter TC, Favoreel HW. 2020. The Attenuated Pseudorabies Virus Vaccine Strain Bartha K61: A Brief Review on the Knowledge Gathered during 60 Years of Research. *Pathogens* 9, 897. (doi:10.3390/pathogens9110897)
5. Evavold CL, Ruan J, Tan Y, Xia S, Wu H, Kagan JC. 2018. The Pore-Forming Protein Gasdermin D Regulates Interleukin-1 Secretion from Living Macrophages. *Immunity* 48, 35-44. (doi: 10.1016/j.immuni.2017.11.013)
6. Fan S, Yuan H, Liu L, Li H, Guan H. 2020. Pseudorabies virus encephalitis in humans: a case series study. *J NeuroVirol* 26, 556-564. (doi: 10.1007/s13365-020-00855-y)
7. Fink SL, Cookson BT. 2005. Apoptosis, Pyroptosis, and Necrosis: Mechanistic Description of Dead and Dying Eukaryotic Cells. *Infect Immun* 73, 1907-1916. (doi: 10.1128/IAI.73.4.1907-1916.2005)
8. He WT, Wan H, Hu L, Chen P, Wang X, Huang Z, Yang ZH. 2015. Gasdermin D is an executor of pyroptosis and required for interleukin-1 β secretion. *Cell Res* 25, 1285-1298. (doi: 10.1038/cr.2015.139)
9. Jorgensen I, Rayamajhi M, Miao EA. 2017. Programmed cell death as a defence against infection. *Nat Rev Immunol* 17, 151-164. (doi: 10.1038/nri.2016.147)
10. Klopfeisch R, Teifke JP, Fuchs W, Kopp M, Klupp BG, Mettenleiter TC. 2004. Influence of tegument proteins of pseudorabies virus on neuroinvasion and transneuronal spread in the nervous system of adult mice after intranasal inoculation. *J Virol* 78, 2956-2966. (doi: 10.1111/j.1600-079X.2008.00631.x)
11. Klopfeisch R, Klupp BG, Fuchs W, Kopp M, Teifke JP, Mettenleiter TC. 2006. Influence of Pseudorabies Virus Proteins on Neuroinvasion and Neurovirulence in Mice. *J Virol* 80, 36-36. (doi: 10.1128/JVI.02589-05)
12. Kogut MH, Lee A, Santin E. 2020. Microbiome and pathogen interaction with the immune system. *Poultry Sci* 99, 1906-1913. (doi: 10.1016/j.psj.2019.12.011)
13. Mcnab F, Mayer-Barber K, Sher A, Wack A, O'Garra A. 2015. Type I interferons in infectious disease. *Nat Rev Immunol* 15, 87-103. (doi: 10.1038/nri3787)
14. Mettenleiter TC 2020. Aujeszky's Disease and the Development of the Marker/DIVA Vaccination Concept. *Pathogens* 9, 563-569. (doi: 10.3390/pathogens9070563)
15. Miller LC, Zanella EL, Waters WR, Lager KM. 2010. Cytokine Protein Expression Levels in Tracheobronchial Lymph Node Homogenates of Pigs Infected with Pseudorabies Virus. *Clin Vaccine Immunol* 17, 728-734. (doi: 10.1128/CVI.00485-09)
16. Pomeranz LE, Reynolds AE, Hengartner 2005. Molecular biology of pseudorabies virus: impact on neurovirology and veterinary medicine. *Microbiology & Molecular Biology Reviews: MMBR* 69, 462-500. (doi: 10.1128/MMBR.69.3.462-500.2005)
17. Romero CH, Meade PN, Shultz JE, Chung HY, Lollis G. 2015. Venereal transmission of pseudorabies viruses indigenous to feral swine. *J Wildlife Dis* 37, 289-296. (doi: 10.7589/0090-3558-37.2.289)
18. Takeuchi O, Akira S. 2010a. Innate immunity to virus infection. *Immunol Rev* 227, 75-86. (doi:10.1111/j.1600-065X.2008.00737.x)
19. Takeuchi O, Akira S. 2010b. Pattern recognition receptors and inflammation. *Cell* 140, 805-820. (doi: 10.1016/j.cell.2010.01.022)
20. Tisoncik JR, Korth MJ, Simmons CP, Farrar J, Katze MG. 2012. Into the Eye of the Cytokine Storm. *Microbiology & Molecular Biology Reviews MMBR* 76, 16-32. (doi: 10.1128/MMBR.05015-11)
21. Wei J, Ma Y, Wang L, Chi X, Yan R, Wang S, Li X, Chen X, Shao W, Chen JL. 2017. Alpha/beta interferon receptor deficiency in mice significantly enhances susceptibility of the animals to pseudorabies virus infection. *Vet Microbiol* 203, 234-244. (doi: 10.1016/j.vetmic.2017.03.022)
22. Wu R, Bai C, Sun J, Chang S, Zhang X. 2013. Emergence of virulent pseudorabies virus infection in Northern China. *J Vet Sci* 14, 363-365. (doi: 10.4142/jvs.2013.14.3.363)
23. Yeh CJ, Lin PY, Liao MH, Liu HJ, Lee JW, Chiu SJ, Hsu HY, Shih WL. 2008. TNF- α mediates pseudorabies virus-induced apoptosis via the activation of p38 MAPK and JNK/SAPK signaling. *Virology* 381, 55-66. (doi: 10.1016/j.viro.2008.08.023)
24. Zhou Z, He H, Wang K, Shi X, Shao F. 2020. Granzyme A from cytotoxic lymphocytes cleaves GSDMB to trigger pyroptosis in target cells. *Science* 368, eaaz7548. (doi: 10.1126/science.aaz7548)
25. Liu SS, Sun W, Huang XF, Yuan R, Yu JS. 2021. Data from: Cytokine storms and Pyroptosis are primarily responsible for the rapid death of mice infected with Pseudorabies virus (doi.org/10.5061/dryad.zw3r2287j)

377
378
379

Figure captions

Figure 1. Histopathological changes of brain of PRV-inoculated mice examined by H.E staining. (A) Microscopic lesion in the control group; (B-E) PRV-infected mice from 24, 48, 60 and 72 hpi.

380
381
382

Figure 2. The dynamic changes of mRNA expression related to immune induced by PRV. (A) *IFN-α*; (B) *IFN-β*; (C) *IFN-γ*; (D) *TNF-α*; (E) *IL-1β*; (F) *IL-6*; (G) *IL-18*; (H) *IL-4*; (I) *IL-10*

Figure 3. PRV caused pyroptosis in mice brain measured by flow cytometry. (A) Pre-infection; (B) 0 hpi; (C) 36 hpi; (D) 48 hpi; (E) 60 hpi; (F) 72 hpi;

Figure 4. In situ expression of caspase-1, IL-1 β and IL-18 in mice brain. (A) Caspase-1 expression in the pre-infection group; (B-D) Caspase-1 expression in brain at 36, 60 and 72 hpi; (E) IL-1 β expression in the pre-infection group; (F-H) IL-1 β expression in brain at 36, 60 and 72 hpi; (I) IL-18 expression in the pre-infection group; (J-L) IL-18 expression in brain at 36, 60 and 72 hpi; Scale bar=50 μ m.

Figure 5. The protein expression related to pyroptosis induced by PRV. (A) The relative amount of protein to β -actin; (B) protein expression related to pyroptosis measured by western blotting.

Figure 1. Histopathological changes of brain of PRV-inoculated examined by H.E staining. (A) Microscopic lesion in the control group; (B-E) PRV-infected mice from 24, 48, 60 and 72 hpi.

119x61mm (300 x 300 DPI)

Figure 2. The dynamic changes of mRNA expression related to immune induced by PRV. (A) IFN- α ; (B) IFN- β ; (C) IFN- γ ; (D) TNF- α ; (E) IL-1 β ; (F) IL-6; (G) IL-4; (H) IL-10

250x181mm (300 x 300 DPI)

Figure 3. PRV caused pyroptosis in mice brain measured by flow cytometry. (A) Pre-infection; (B) 0 hpi; (C) 36 hpi; (D) 48 hpi; (E) 60 hpi; (F) 72 hpi;

99x65mm (300 x 300 DPI)

25 Figure 4. In situ expression of caspase-1, IL-1 β and IL-18 in mice brain. (A) Caspase-1 expression in the
26 pre-infection group; (B-D) Caspase-1 expression in brain at 36, 60 and 72 hpi; (E) IL-1 β expression in the
27 pre-infection group; (F-H) IL-1 β expression in brain at 36, 60 and 72 hpi; (I) IL-18 expression in the
28 pre-infection group; (J-L) IL-18 expression in brain at 36, 60 and 72 hpi; Scale bar=50 μ m.

29 199x112mm (300 x 300 DPI)

Figure 5. The protein expression related to pyroptosis induced by PRV. (A) The relative amount of protein to β -actin; (B) protein expression related to pyroptosis measured by western blotting.

150x61mm (300 x 300 DPI)

Appendix B

Dear editors and reviews,

Thank you for your kind comments on our manuscript entitled "Cytokine storms and pyroptosis are primarily responsible for the rapid death of mice infected with pseudorabies virus". Those comments are all valuable and very helpful for revising and improving our paper. We have studied the comments carefully and have revised the manuscript according to the reviews' comments and suggestions. Revised portions are marked in red in the paper.

Responses to the editors:

All wrong spelling has been revised in the manuscript.

(1) Line 1 storms

Response: we have revised the line 1, 20, 135, 169, 186, 206 and 219 storms.

(2) Line 21 respectively.

Response: we have revised the line 20 : respectively.

(3) Line 32 largely

Response: we have revised the line 30 : largely.

(4) Line 38 virulence

Response: we have revised the line 36 : virulent.

(5) Line 39 basing

Response: we have revised the line 37 : based.

(6) Line 45 It known

Response: we have revised the line 37 : It is known.

(7) Line 48 proinflammatory

Response: we have revised the line 47 : pro-inflammatory.

(8) Line 52 pyroptosis-related

Response: we have revised the line 51: pyroptosis-related.

(9) Line 62 Eighty

Response: we have revised the line 62: Eighty.

(10) Line 75 photophate

Response: we have revised the line 75: phosphate .

(11) Line 87 visualized

Response: we have revised the line 87: visualize.

(12) Line 95 were

Response: we have revised the line 96: was.

(13) Line 97 corrction.

Response: we have revised the line 97: correction.

(14) Line 101 bradford

Response: we have revised the line 101: Bradford.

(15) Lines 110-111: SPSS22.0 software was used to data analyses. The results were espresented as means \pm standards deviations(mean \pm SD)

Response: we have revised the line 110-111: SPSS22.0 software was used for data analyses. The results were presented as means \pm standards deviations (mean \pm SD).

(16) Line 150, 197 tredency

Response: we have revised the line 153 and 203: tendency.

(17) Line 151 manitained

Response: we have revised the line 154: maintained.

(18) Line 152 compareing and 161 comparint

Response: we have revised the line 155: comparing it, and 162 compared.

(19) Line 154 levels

Response: we have revised the line 157: levels.

(20) Line 164 belong

Response: we have revised the line 169: belongs.

(21) Line 205 consist

Response: we have revised the line 212: consistent.

(22) Line 226 wirk

Response: we have revised the line 229: work.

Responds to the reviews' comments:

Reviewer: 1

Comments to the Author(s)

This paper has some merit addressing an important issue about the role of the immune response in the pathogenesis of the ADV infection in the mouse model.

(1) However, it needs a profound revision on the language, more detail in the figure legends, and a complete description of the experimental design (groups and number of animals).

Response: we have revised thought the manuscript to enhance the expression, grammar constructions and word spelling.

Figure legends have been improved in figure 1-5 in lines: 387-420.

Eighty 6-week-old female Balb/C mice were used in this study. We have revised in lines 64-65: “One week later, the mice were divided randomly into five groups with sixteen per group, including one control group and four experimental groups, which mice were infected at 36 hpi, 48 hpi, 60 hpi and 72 hpi, respectively.”

(2) Also I would recommend to discuss the evident limitations of the mouse model when extrapolating to economical species suffering of Pseudorabies.

Response: We have revised in lines 216-218: “Although cytokine storms and pyroptosis might be the cause for the rapid death of mice caused by PRV strain. However, due to the difference of immune system between mice and pig, more detail information about the pathogenesis to pig and other mammal animals need to be further clarify in future.”

(3) The statistical results also should be more explicit in the text and figures to strengthen the discussion.

Response: We have revised in “3.2 The mRNA expression levels about innate immune-related genes” from line 125-142.

Figure 1-4 have been improved to strengthen the results such as marking the statistical difference, and clarifying the the results and discussion sections.

(4) Going through the experimental design, it is not clear how the groups of animals were conformed. A total of eighty mice were divided into two groups (probably 40 control and 40 experimental), however, 6 animals of each group were killed at 4 different time points, given a total of 24 mice per group, what happened with the rest of the animals?

Response: We have revised in lines 64-65 to clarify the animal groups in this work: “One week later, the mice were divided randomly into five groups with sixteen per group, including one control group and four experimental groups, which mice were infected at 36 hpi, 48 hpi, 60 hpi and 72 hpi, respectively.”

(5) On figure 2 six time points were described (36 mice), however, no results on the control group were shown. All figures need a self-explanatory text. Figure 2 misses IL-18 text.

Response: all statistical results were compared with that in the control group. And figure coordinate has been relabeled so as not to cause misreading.

All figure legend have been modified and IL-18 text in figure 2 has been added in line 398.

(6) Although a statistical analysis is claimed to be applied on the results, differences were not shown in the text or on the figures. Since the groups of animals were not accurately described, the analysis must be detailed.

Response: we have added the statistical difference in the results of manuscript and figure 2.

The animals used in this study has been clearly described in the line 64-65. And the detail information about the results and discussion has been revised in the manuscript.

The other sections revised have been marked in red in the manuscript.

Thanks again for the excellent and professional revision of our manuscript. Hopefully, we could have our article been considered of publication in this journal. Should there been any other corrections we could make, please feel free to contact us by email. My email is sunwei_223@163.com.

Yours sincerely,

Wei Sun

July 18th, 2021

Appendix C

Dear editors and reviews,

Thank you for your kind comments on our manuscript entitled "Cytokine storms and pyroptosis are primarily responsible for the rapid death of mice infected with pseudorabies virus". Those comments are all valuable and very helpful for revising and improving our paper. We have studied the comments carefully and have revised the manuscript according to the reviews' comments and suggestions. Revised portions are marked in red in the paper.

Responds to the reviews' and editors' comments:

Reviewer: 1

Comments to the Author(s)

This paper is now suitable for publication. However, requires a further review of the language to make it more understandable.

Response: With the help of INCRESCIENCE co., Ltd (<https://check.newacademic.net>, a service resources purchased by our college), we have made extensive amendments on the manuscript to improve clarity, enhance expression and grammar constructions provide by the helping. We believe this revised manuscript is greatly improved in language use.

The detailed modification information is as follows:

Category	Score
Plagiarism	67 of 100
Grammar	103
Contextual Spelling	10
Misspelled words	28
Unfamiliar words	6
Confused words	3
Word clusters of English	1
Grammar	30
Determiner use (a/an/the/this, etc.)	6
Faulty subject-verb agreement	6
Incorrect noun number	4
Wrong or missing prepositions	4
Punctuation	10
Comma misuse within clauses	11
Punctuation in compound/complex sentences	4
Misuse of semicolons, quotation marks, etc.	1
Sentence Structure	8
Unrelated sentences	2
Misplaced words or phrases	1
Style	11
Wordy sentences	6
Improper formatting	1
Unclear text	2
Repetitive issues	2
Vocabulary enhancement	8
Word choice	4

*Author for correspondence: liuyangli@caas.cn
liuyangli1@caas.cn

† These authors contributed equally to the work
Present address: College of Agriculture, Tongren
Polytechnic College, Wuyang District, Tongren City,
Guizhou, 550000, China

Introduction

Parasitoid virus (PV) is said to be present in the pathogens of porcine Leishmaniasis disease (ALD), which causes the respiratory system, nervous system and reproductive system [1]. Many mammals, including pigs, are susceptible to PV infections, such as cattle, sheep, rabbits, cats, dogs, guinea pigs, rats and mice [2]. However, pigs are the only susceptible animals that can survive, although the prognosis of the disease largely depends on the factors including weathering site, virus strain and time of infection [3]. ALD is a highly infectious disease with high mortality in piglets. Transmission mainly occurs through direct contact with oral and nasal secretions but can also occur through aerosol and the placenta or sexual intercourse [4]. Therefore, the persistence of PV has led to a wide range of economic losses in the pork production industry. Inactivated and attenuated vaccines have been developed to delay or reduce viral death. However, they cannot eradicate the disease because most of them can prevent viral penetration and reproduction and shedding of viruses [5-8]. Due to the impact of ALD on the pig industry, some countries are trying to eradicate ALD based on the DIVA (Differentiating Infected From Non-infected animals) program. However, since 2011, the outbreak of ALD occurred in pigs vaccinated with PR vaccine in China, which indicates that the ALD vaccine can not provide effective protection to prevent viral infection [9].

Mice and calves are equally used to study PV in the laboratory. After infection, animals showed abnormal reactions and nasal discharges, accompanied by prostration and rapid death. In mice, PV always manifested as a

respiratory infection of the central nervous system (CNS), accompanied by fibrinogen central nervous system and high mortality [10]. PV is known to cause severe encephalitis in piglets, various neurologic lesions, even to humans [11]. Few studies have focused on the pathogenesis of encephalitis. It is known that pyroptosis is involved in the immune response to various types of cells, which can be triggered by a variety of pathological stimuli, leading to the secretion of pro-inflammatory cytokines and membrane constructs, inflammation is a double-edged sword, which has a crucial role in resolution. A mild inflammatory response could protect the body to a certain degree, help to repair damaged tissues, and be beneficial to steady-state re-organization. Nevertheless, excessive inflammation may form “cytotoxic storms”, leading to tissue damage. In the present work, we describe the influence of PV on the immune factor and pyroptosis-related factor in mice brain.

Anti-viral-V-FITC/PV Antigen Kit was obtained from BD Company (Franklin Lakes, USA). Modified Bradford Protein Assay Kit, Antibody against NLRP3, Animal Total RNA Isolation Kit and Three-Step Protein Extraction Kit were supplied by Sugan Biotech Company (Shanghai, China). Antibody against caspase-1 (SC44546) was bought from Thermo Fisher Company (USA). IL-1 β , IL-18 and β -actin antibodies were obtained from Boster Company (Beijing, China). Caspase-1 antibody was obtained from Boster Biological Technology Co., Ltd (Wuhan, Hubei, China). Proteinase K (E-caspr) Kit was bought from DANAPA company (Dalian, China). RNase H-L1 stain (R6200276) obtained from Hockling Jiang was provided by Professor English Wang from Harbin Veterinary Research Institute, CAH. Eighty 6-week-old female Balb/c mice were obtained from Dony Experimental Animal Cooperation (Zhengzhou, China).

2.2 Experimental

The wild mice were divided randomly into five groups with eleven per group, including one control group and four experimental groups, which mice were infected at 36 dpi, 48 dpi, 60 dpi and 72 dpi. The mice in the

control group were injected with 0.2 ml of normal saline (NS) by subcutaneous inoculation on the back. The mice in another group were received 0.2 ml PV-NS3 vaccine (104 TCID₅₀ 100 μ l) at the same inoculation site. The animals were fed on the usual fluorescent with a 12 h light-dark cycle. The infection temperature and relative humidity were maintained at 22-24°C and 60-65%, respectively. Ten mice were fed in each cage and given the above dosage. Water and feed were provided ad libitum. Mice brain tissues in each group were collected aseptically. Infection specimens were snap frozen and stored at -80°C for RNA extraction. In addition, portions of the brain were fixed in 4% paraformaldehyde solution for histopathological examination.

2.3 Histopathological analysis

The histopathological observation was operated by using a standard laboratory procedure. The brain was removed from experimental animals and washed thoroughly in phosphate buffered saline (PBS, pH 7.4). Then, the tissue was fixed in 4% paraformaldehyde for 2.0 h, and transferred to 30% sucrose in PBS solution at 4°C for dehydration. After that, processed in parallel embedding machine. The paraffin-embedded tissue was sliced into a five μ m sections, directed to yellow and then rehydrated in graded alcohols. The section was stained with hematoxylin-eosin (H-E) staining and then examined under a light microscope for histopathological examination.

2.4 Detection of pyroptosis by flow cytometry to brain

The rate of pyroptosis cells in the brain was measured using an Annexin V-FITC/PI Apoptosis Kit according to the instructions provided. Brains were taken from mice, which were humanely killed at a time where mentioned, ground to fine a suspension and filtered with a 100-mesh nylon screen. The cells were washed three times with pre-cooled PBS and adjusted at a concentration of 1×10^6 cells/ml. Furthermore, 100 μ l cells were incubated with Annexin V-FITC/PI staining at room temperature for 15 min in a culture tube in a dark atmosphere. Each tube was added with 300 μ l of binding buffer and then filtered with an FCN (Falcon) (Becton Dickinson, USA). CellJet Pro software (Beckon Dickinson, USA) was used to visualize the results.

2.3 RNA extraction and qRT-PCR analysis

Total RNA from brain tissue was isolated by an “Animal RNA Isolation Kit” according to the kit instructions. RNA integrity was detected by 2% agarose gel electrophoresis. A spectrophotometer was used to detect the RNA quantity and quality (NanoDrop-2000, ThermoFisher Company, USA). Total RNA was converted into cDNA by qRT-PCR by using the PrimeScript RT reagent Kit. The first strand of cDNA was amplified through SYBR melting on a LightCycler 96 apparatus (Roche, Germany). All primers used in this research were designed by Clintx 7 software and synthesized by Sugan Biotech Ltd. (Shanghai, China). Detailed information about primers was available in Table S1. GAPDH method was used for the analysis of β -actin expression. β -actin, a housekeeping gene, was used as an internal control for values correction.

2.4 Western blotting

A “Three-Step Protein Extraction Kit” was used to protein extracted from the brain. The protein concentration of each specimen was measured by Bradford assay. The proteins were first separated on a 10% sodium dodecyl sulfate polyacrylamide gel electrophoresis (SDS-PAGE) followed by being transferred onto a polyvinylidene difluoride (PVDF) membrane. Then, the PVDF membrane was blocked with TBST solution contained with 3% bovine milk at room condition for 2 h and then incubated at 4°C condition for 12 h with the corresponding primary antibodies diluted with a solution ALBPS-1 (1:1000), Caspase-1 (1:1000), GSDME (1:1000), IL-1 β (1:1000), IL-18 (1:1000) and β -actin (1:1000). Then, the PVDF membrane was incubated at room temperature with HRP-labeled secondary antibody for 1 h and then were measured by an ECL reagent. The β -actin was used as a protein loading control.

2.7 Statistical analysis

SPSS22.0 software was used for data analysis. The results were presented as mean \pm standard deviation (mean \pm SD). One-way analysis of variance (ANOVA)

perhaps in SPSS 22.0 was used to evaluate the statistical significance between PVV and control group. GraphPad Prism 8.0 software was used to perform statistical analysis. In all statistical comparisons, the p-value was included as a judgement of the statistical difference.

2.Results

2.1 Clinical symptoms and histopathological analysis

At 30 dpi, the mice infected with PVV generally appeared typical clinical signs, including depression, anorexia and emaciated body. Mortality occurred within the period of 40-72 dpi. However, none of the mice in the control group exhibited clinical symptoms or died. Compared with the control group, there was a significant difference in the microscopic lesions in the PVV-infected group (Figure 1). Normal brain structure was found in the control and PVV-infected mice before 30 dpi (Figure 1A and B). At 40 dpi, hippocampus appeared in brain tissue, followed by perivascular space widening, perivascular lymphocyte movement, as well as degeneration and necrosis occurred in some neurons (Figure 1C). Focal inflammatory cellular infiltration was found in the brain at 60-72 dpi (Figure 1D and E).

2.2 The mRNA expression levels of genes related to immune response

Immune response recognizes invading pathogens by binding to pattern recognition receptors, leading to the expression of antiviral molecules. Interferon (IFN) is an antiviral molecule, which has a general role in the clearance of invading pathogens, such as interferon (IFN). In this work, the transcriptional levels of IFN- α , IFN- β and IFN- γ were measured by qRT-PCR. As shown in Figure 2, the expression levels of IFN- α and IFN- β in the brain of PVV-infected mice were up-regulated from 30 dpi ($p < 0.01$) and peaked at 40 dpi ($p < 0.001$) and then down-regulated to 60-72 dpi ($p < 0.05$, Figure 2A-B). Furthermore, IFN- γ expression in brain was down-regulated before 30 dpi and then up-regulated until 60 dpi ($p < 0.01$, Figure 2C). In addition, pro-inflammatory cytokines TNF- α , IL-1 β , IL-6 and IL-17 and anti-inflammatory cytokines IL-4, IL-10 were also measured in the brain, respectively. The relative mRNA

expression levels of IL-1 β and IL-17, two cytokines related to pyroptosis, were determined by blotting. The results demonstrated that PVV could elevate the protein levels of IL-1 β and IL-17.

2.3 Relative mRNA expression of genes related to pyroptosis in the brain

The mRNA expression levels of pyroptosis-related factors were further detected by qRT-PCR in this research. As shown in Table S2, the mRNA expression level of NLRP3 was up-regulated ($p < 0.01$) in a time-dependent manner following PVV infection. Moreover, the mRNA expression levels of Caspase-1 were also up-regulated ($p < 0.01$) when compared with the control group. Besides, the GSDMD mRNA expression level was up-regulated in an increasing tendency ($p < 0.01$) compared with that in the control group.

3.Discussion

PVV is a kind of non-enveloped β herpesvirus, which belongs to the genus Varicellovirinae, family Herpesviridae. Pigs are the only natural hosts of PVV for their natural infection. However, mice and rats can be naturally infected with PVV and cause a fatal disease. In the lab, also natural infection in adult mice, PVV causes perivascular nerve cells and spreads to the central nervous system [6]. Previous studies on PVV mainly focused on pathogenicity and the resulting host immune response. The aim of this research was to describe the kinetics of cytokine secretion *in vivo* and to clarify whether cytokine storm is involved in the pathogenesis of PVV. In this work, we identified cytokine storm and pyroptosis as the main causes of rapid death in mice infected with PVV.

The immune system is the first line in defending against the invasion of pathogens, accompanied by the recruitment of immune cells and inflammatory responses [1]. Inflammation is a process to solve microbial infection and a complex process involving the regulation of cytokine production. Dysfunction of these factors can induce cytokine storm and related multiple organ failure [2]. An inflammatory response is usually caused by various of pro-inflammatory cytokines, such as TNF- α , IL-1 and IL-6. These cytokines are the kind of pleiotropic

levels of TNF- α and IL-6 in the brain were remarkably up-regulated caused by PVV infection and peaked at 40 dpi ($p < 0.01$) followed by down-regulated at 72 dpi (Figure 2 D and E). After PVV infection, the IL-1 β expression in the brain was up-regulated from 30 dpi and lasted for the whole experiment period ($p < 0.01$, Figure 2E). In addition, IL-1 β expression was up-regulated from 30 dpi and had the same tendency as IL-6 ($p < 0.01$, Figure 2E). The cytokine storm caused by IFN- α , IFN- β , TNF- α , IL-1 β , IL-6 and IL-17 were related to the histopathological changes induced by PVV (Figure 1). Notably, the mRNA expression levels of IL-4 and IL-10 were up-regulated before 40 dpi ($p < 0.01$) and then down-regulated and maintained a low level until 72 dpi (Figure 2I and J). This indicated that PVV inhibited the expression of IL-4 and IL-10 since 40 dpi.

3.1 Detection of pyroptotic cells in the brain

Pyroptosis in the brain was measured by Annexin V-FITC double staining through flow cytometry. As shown in Figure 3, a large number of PI⁺ pyroptotic cells in the brain was observed at 30, 40, 60 and 72 dpi, which increased in a time-dependent way ($p < 0.01$) compared with that in the control group (Table S3). In addition, PVV significantly up-regulated to increase the protein expression of caspase-1 (Figure 4D-E), IL-1 β (Figure 4F-H) and IL-17 (Figure 4I-L) in the brain tissue in a time-dependent manner.

3.2 Changes in protein expression levels related to pyroptosis in the brain

To further investigate the effect of PVV on the pyroptosis *in vivo*, the proteins involved in pyroptosis - signal pathway were detected by blotting, including NLRP3, Caspase-1, GSDMD, IL-1 β and IL-17. From the result in Figure 3, NLRP3 decreased at 30 dpi and increased in an increasing tendency in a time-dependent manner. In addition, the Caspase-1 level was significantly up-regulated by PVV from 30 dpi and maintained at a high level until the end of this experiment by comparing it with that in the control group. Furthermore, GSDMD, a pyroptosis marker protein, was also increased in all groups. As shown in Figure 5A, the amount of GSDMD was higher in all PVV groups than that in the control. In addition, the

proteins involved in the regulation of cell death in inflammatory disease, vascular endothelial cell permeability, attracting the blood cells to inflammatory tissue, and acute phase - protein production [7].

Cytokines play an important role in all forms of the immune response, coordinating the innate and adaptive immune responses. Consequently, in most cases, cytokines play a greater role in initiating immunogenic and regulatory reactions, such as tissue injury and microbial invasion. IFN- α are recognized as the central factors of antiviral infection, which has a general role in immune defense response [8]. In addition, cytokines and interleukin play an important role in the pathogenesis of antiviral and viral infection. However, excessive immune activation and excessive release of cytokines could be rather detrimental [9]. For example, over-expression of TNF- α , IL-1, and IL-6 in the immune system could lead to vascular leakage, systemic fatigue, cardiomyopathy, and acute phase - protein synthesis [10]. In addition, persistent excessive IFN- α may also be harmful to the immune system [10]. In the present study, we found that cytokine storm was induced by PVV in mice brain from 30-72 dpi including the elevated expression levels of Type I IFNs (IFN- α and IFN- β) and Type II IFN (IFN- γ) as well as pro-inflammatory factors compared to that in the control group ($p < 0.01$). This result was consistent with the histopathological changes including hippocampus and inflammatory cell infiltration in the mice brain. Studies have shown that PVV could regulate the expression of cytokines, including type I and type II interferon and inflammatory factors, to establish a successful infection [16,17]. Furthermore, IFN- α and IFN- β mediate a positive feedback regulation by binding to IFN- α and IFN- β receptor as an autocrine or paracrine manner [18]. PVV infection can induce apoptosis, which has been reported previously *in vitro* and *in vivo* [19,20]. However, apoptosis is usually considered as an inducible programmed cell death (PCD), which is characterized by an active programmed process of cell decomposition to avoid inflammation [21]. The discrepancy was found in this research may be induced by a new kind of PCD in the cell

death process. Pyroptosis is a new type of pro-inflammatory cell death, which is emerging as the mechanism of unsparring and clearing pathogen infection and requires the activation of caspase-1 (45)(122). In this research, an increasing number of pyroptosis cells induced by PIV were observed in the brain through flow cytometry by comparing with the control group (p=0.01). Furthermore, the expression of genes and proteins related to pyroptosis pathways were observed by qPCR and Western blotting methods, respectively. Inflammation and pyroptosis are closely related to central nervous system (23). Once activated, caspase-1 can process and mature IL-1 β and IL-18 precursors, as well as cleave GSDME, resulting in cell membrane channel opening and the pyroptosis (45). Among the inflammasomes, NLRP1 is currently the most well known one, which response to various stimuli. NLRP1 was activated by PIV in this research. In addition, GSDME, a key molecule in pyroptosis, also was activated by PIV. The activity of GSDME leads to the indirect release of IL-1 β and IL-18 from membrane pores (24). The pyroptosis cell death pathway provides a large amount of inflammatory response at the site of infection. This is consistent with the results from histopathologic analysis (Figure 1) and immunohistochemistry (Figure 4), as well as a full explanation for cytokine storm caused by PIV in mice brain. Although cytokine storm and pyroptosis might be the cause for the rapid death of mice caused by PIV virus. However, due to the difference of immune system between mice and pig, more detailed information about the pathogenesis in pig and other mammal animals need to be further clarified in the future.

Please check up the comments made on the yellow marked words or phrases, as well as on the stroke out words in the reviewed PDF file and be sure to amend the errors.

(1) Line 17 PRV

Response: Pseudorabies virus can be abbreviated as PRV or PrV according the references.

(1) Line 22 immune

Response: we have revised in line 19 “ immune reaction”.

(2) Line 24 a

Response: we have deleted “a” line 21.

(3) Line 31 Besides

Response: we have deleted “a” line 28 “Many mammals, including pigs,”.

(4) Line 34 et al

Response: we have deleted “a” line 31 “.....as well as the age of pigs”.

(5) Line 36 through

Response: we have revised in line 33 “and”

(6) Line 67 respectively

Response: we have deleted “respectively” line 65.

(7) Line 69...was...of...

Response: we have revised in line 67 “...were received...”

(8) Line 84 protocol

Response: we have revised in line 82 “...instructions provided”.

(9) Line 92 directions

Response: we have revised in line 90 “... instructions”.

(10) Line 117 showed

Response: we have revised in line 116 “...appeared...”.

(11) Line 131 up

Response: we have revised in line 130 “...down-regulated...”.

(12) Line 136 but maintain at a high level until 72 hpi (p<0.01 and p<0.05, Figure 2D and F)

Response: we have revised in line 135 “...by down-regulated at 72 hpi (Figure 2 D and F)”.

(13) Line 138 However

Response: we have revised in line 136-137 “In addition...”.

(14) Line 143 pyroplic

Response: we have revised in line 142 “...pyroptosis...”.

(15) Line 146 than

Response: we have revised in line 145 “...that...”.

(16) Line 152 Caspae-1

Response: we have revised in line 151 “Caspase-1”.

(17) Line 166 *Varicelloviru*

Response: we have revised in line 168 “*Varicellovirus*”.

(18) Lines 212-213 detail information about the

Response: we have revised in line 215-216 “detailed information about the pathogenesis to pig and other mammal animals need to be further clarified in the future”.

(19) Lines 152, 158 and 202 western

Response: we have deleted western in lines 151, 157 and 204.

The other sections revised have been marked in red in the manuscript.

Thanks again for the excellent and professional revision of our manuscript. Hopefully, we could have our article been considered of publication in this journal. Should there been any other corrections we could make, please feel free to contact us by email. My email is sunwei_223@163.com.

Yours sincerely,

Wei Sun